# Effect of MgCl_2_ Loading on the Yield and Performance of Cabbage-Based Biochar

**DOI:** 10.3390/bioengineering10070836

**Published:** 2023-07-15

**Authors:** Cui Zhu, Kuncheng Huang, Mengyuan Xue, Yiming Zhang, Jiaquan Wang, Lu Liu

**Affiliations:** 1School of Mechanical Engineering, Hefei University of Technology, Hefei 230009, China; xihuangdongling@163.com (C.Z.); huangkuncheng999@163.com (K.H.); 2Anhui Jincheng Anhuan Technology Development Co., Ltd., Hefei 230000, China; 3Anhui Institute of Quality and Standardization, Hefei 230002, China; 4School of Engineering, Anhui Agricultural University, Hefei 230036, China; xuexxx@stu.ahau.edu.cn (M.X.); zhang1001@stu.ahau.edu.cn (Y.Z.)

**Keywords:** cabbages, biochar, magnesium chloride, carbon yield, fixed carbon yield

## Abstract

Converting more CO_2_ absorbed by plant photosynthesis into biomass-activated carbon effectively reduces carbon emissions. In this study, we used a one-step preparation of biomass-activated carbon loaded with MgO nanoparticles to investigate the effect of Mg loading on the catalytic pyrolysis process. The influences of magnesium loading on biochar yield and fixed carbon production were assessed. The addition of 1% Mg weakened the carbonyl C=O, inhibited the dehydroxylation reaction, enhanced the C-H signal strength, and the formation of MgO inhibited the weaker- bound substituent breakage. Additionally, the addition of magnesium altered the morphological features and chemical composition of the biochar material. It also increased the activated carbon mesoporosity by 3.94%, biochar yield by 5.55%, and fixed carbon yield by 12.14%. The addition of 1% Mg increased the adsorption capacity of the activated carbon to potassium dichromate, acid magenta, methylene blue, and tetracycline effluents by 8.71 mg, 37.15 mg, 117.68 mg, and 3.53 mg, respectively. The results showed that MgCl_2_ played a significant role in promoting the thermal degradation of biomass and improving the solid yield and adsorption performance of activated carbon.

## 1. Introduction

The energy demand has increased drastically owing to the sharp growth in the world’s population and rapid industrial development, both of which have increased consumption of the world’s already limited energy reserves and posed severe environmental challenges [1,2]. In this context, we need sustainable and environmentally friendly alternative energy sources to reduce the carbon footprint of energy generation. The utilization of biomass energy has attracted significant attention to solving the energy shortage due to the abundance of raw materials [3,4,5,6].

Charring is the primary thermochemical treatment of biomass and is an efficient method to produce biochar by pyrolysis of biomass [7,8,9]. Biochar mainly comprises aromatic hydrocarbons with a carbon content of about 60%. Because of its high fixed carbon content, biochar has the potential to replace conventional fossil fuels [10,11]. Therefore, biochar is an essential product of biomass. The biomass char reduction in atmospheric CO_2_ gas modelled by Lehmann et al. [12] concluded that biomass carbonization is also a practical carbon sink pathway.

Biochar is an essential product of biomass, so studying carbonization conditions to improve biochar’s yield and fixed carbon content has great scientific significance and application. During carbonization, the most critical factors that can be easily controlled are temperature, heating rate, and holding time [13,14]. All of them greatly influence the quality and yield of the final product [15]. Wei et al. analyzed the effect of temperature on the properties of biochar from different raw materials [16,17,18,19]. Mohamed et al. analyzed the effect of holding time on the yield of biochar [20]. Wannapeera et al. studied the effect of temperature and holding time on biochar properties [21,22,23]. Safdari and Zanzi et al. investigated the effect of heating rate and temperature on yield, respectively [24,25]. In biochar studies, yield is an essential indicator [26], and fixed carbon content is an important quality indicator. The carbon emission can be reduced by adding a small amount of carbon sequestration agent. In that case, it will have great significance for ecological environment protection. The loading of MgCl_2_ can promote the thermal degradation process of pyrolysis, weaken the hydrogen bond network, destroy the crystal structure, and enhance the crosslinking and repolymerization reaction of pyrolysis intermediates. At the same time, the MgO particles generated by pyrolysis are embedded between the elastic layers, increasing the specific surface area and voids of activated carbon [27]. The study which improved the yield and fixed carbon content of biochar by adding MgCl_2_ carbon fixing agent has been less than the study which changed the carbonation conditions. Natural seawater is rich in a high concentration of MgCl_2_, which can be easily absorbed into the biomass matrix as Mg salt precursors. Activated carbon is the most effective adsorbent for removing contaminants from water. However, the application of large amounts of commercially available coal-based activated carbon in large-scale water treatment processes may pose serious problems from an economic point of view.

In this paper, the effect of MgCl_2_ loading on cabbage-based high-performance biochar materials and the mechanism of the catalytic pyrolysis process was investigated. The thermal degradation process behavior and catalytic pyrolysis experiments were investigated using a thermogravimetric analyzer (TGA). In addition, the physical and chemical properties of biochar were analyzed and characterized in-depth. The effect of MgO-loaded activated carbon materials on the adsorption performance of pollutants was investigated. To achieve high-quality utilization of biomass resources and reduce carbon emissions, we think that improving the yield of biomass-activated carbon using Mg loading is a promising method.

## 2. Materials and Methods

### 2.1. Materials

Cabbage waste was derived from the vegetable market in Hefei, China. ZnCl_2_, methylene blue, acid magenta, tetracycline, potassium dichromate, and MgCl_2_ were purchased from Sinopharm Chemical Reagent Co., Ltd. (Shanghai, China).

### 2.2. Preparation of Carbon Materials

Cut 70 g of fresh cabbage waste into 0.5 cm long square pieces (about 96.45% water content). Add 15 g of zinc chloride and 0.25 g of magnesium chloride, stir well, and wait for the cabbage to begin to dehydrate. Then, transfer the dehydrated cabbage into 100 mL of a hydrothermal kettle. Finally, close the hydrothermal kettle and place it in a 200 °C oven. Keep for 24 h. Then, remove the hydrothermal kettle. Let it cool to room temperature. The mixture was transferred to a beaker and heated to 200 °C for 24 h for reaction drying. The samples were prepared and labelled as 1% Mg-ZnCHC. Without adding MgCl_2_, it was labelled ZnCHC, and the sample with only cabbage was labelled CHC. The U-shaped quartz tube holding 2 g of hydrothermal carbon was quickly put into the heating zone and maintained for 60 min before being removed from the tube furnace and allowed to cool to room temperature. Throughout the process, the quartz tube was continuously filled with nitrogen at a flow rate of 30 mL/min. The pyrolysis samples were gathered and then rinsed in boiling water until the pH was about 7. After drying, the samples were named 1% Mg-ZnCC, ZnCC, and CC.

### 2.3. Thermogravimetric Test

Samples labelled as 1% Mg-ZnCHC, ZnCHC, and CHC were subjected to pyrolysis tests using a thermogravimetric analyzer (TGA8000, Perkin Elmer, Waltham, MA, USA). In an alumina crucible, a 10 mg sample was heated to 600 °C at a ramp rate of 10 °C/min while being exposed to a nitrogen flow rate of 100 mL/min in order to lessen the impact of buoyancy and establish a stable baseline, a blank test was carried out for each sample measurement.

### 2.4. Characterization of Activated Carbon

Scanning electron microscopy (SEM, Regulus 8230, Hitachi, Tokyo, Japan) and field emission transmission electron microscopy (TEM, JEM1400FLASH, JEOL, Tokyo, Japan) were used to determine the morphology of the materials. The nitrogen adsorption-desorption isotherm (Autosorb-IQ 3, Quanta Chrome, Boynton Beach, FL, USA) was used to determine the structural properties of the materials. The pore volume was calculated using the t-plot method, and P/P0 ranged from 0.2 to 0.5. The chemical states of elements and functional groups in samples were analyzed by X-ray photoelectron spectroscopy (XPS, ESCALAB 250 Xi, Thermo, Waltham, MA, USA). The materials’ chemical makeup was assessed using Fourier transform infrared spectroscopy (FTIR, Nicolet iz10, Thermo Fisher Scientific, USA). The technique utilized was x-ray diffraction (XRD, MAX2500VL, Ritani, Tokyo, Japan). The zeta potential of the materials was measured using a zeta potential analyzer Nano ZS90 (China).

### 2.5. Adsorption Experiment

For batch adsorption tests, 10 mg of 1% Mg-ZnCC and ZnCC were accurately weighed and placed in 250 mL conical bottles. They were then added to 50 mL of methylene blue solution, acidic magenta solution, potassium dichromate solution, and tetracycline solution with an initial concentration of 100 mg·L^−1^ (pH = 7.0), respectively. The adsorption experiments were performed in a shaker at 180 rpm for 1440 min (T = 298 K). Following adsorption equilibrium, UV-V spectrometry was used to measure the concentrations of the solution at 664 nm, 540 nm, 275 nm, and 356 nm, respectively.

## 3. Result and Discussion

### 3.1. Effect of MgCl_2_ Loading on the Thermal Degradation Process

The three distinct peaks in the DTG curve of CHC can be seen in Figure 1b, which shows that pyrolysis can occur in three stages. The first stage was when the temperature was raised from 30 °C to 170 °C. In this stage, CHC loses about 5% of its weight primarily due to the evaporation of free water. The mass loss of CHC was evident in the second stage when the temperature was raised from 170 °C to 250 °C, indicating that rapid pyrolysis of cabbage occurred in this stage, likely due to the decomposition of organic substances in CHC. The corresponding DTG curves showed more significant weight loss at 247 °C and 310 °C, and these two weight loss peaks resulted from hemicellulose and cellulose decomposition, respectively. This is in line with previous research on the pyrolysis temperatures of hemicellulose, cellulose, and lignin, which range from 200 to 280 °C for hemicellulose, 240 to 350 °C for cellulose, and 280 to 500 °C for lignin, the three components of biomass [28]. It is clear that the temperature at which hemicellulose pyrolyzes is low and the temperature at which lignin pyrolyzes is very high. The majority of cellulose and hemicellulose in the lower temperature range is pyrolyzed into molecules with smaller molecular weight and volatilized out in the gaseous state. On the other hand, lignin decomposes slowly over a more comprehensive temperature range, which promotes the production of biochar and lessens the condensation of aromatic carbon functional groups [29]. Hemicellulose, cellulose, and lignin in CHC are all pyrolyzed in the third stage once the temperature hits 500 °C, leaving only the aromatic and unsaturated hydrocarbon compounds in the char for the thick cyclization coking reaction [30].

The addition of ZnCl_2_ alters the way that cabbage is pyrolyzed, as shown by the TG-DTG curves of ZnCHC and 1% Mg-ZnCHC in Figure 1. Three peaks are more visible from the DTG curves, indicating that the hydrothermal carbon pyrolysis of cabbage with the addition of ZnCl_2_ also has three processes. Before 150 °C, the first peak of the mixture DTG curve appears in advance to about 110 °C, which was 60 °C lower than CW. The second peak was 189 °C, which was roughly 60 °C lower than CW. The addition of ZnCl_2_ lowered the pyrolysis temperature of cabbage, which was favorable to the raw material’s rapid pyrolysis, as evidenced by the lower temperature of the interval between the peaks of the DTG curve in the low-temperature zone. After 600 °C, a significant weight loss peak is mainly brought on by ZnCl_2_ gradually melting into the liquid phase [31]. When ZnCl_2_ is added, the temperature at which cabbage decomposes decreases, which can lower the activation energy of cabbage pyrolysis. By comparing the DTG curves of the ZnCHC and 1% Mg-ZnCHC samples, it can be seen that adding 1% Mg can further lower the material’s pyrolysis temperature and reduce the weight loss rate to around 190 °C. The decomposition of hemicellulose and cellulose into gaseous volatiles of smaller molecular weight was reduced by blocking the rate of pyrolysis to reduce the release of carbon-containing gaseous material, and this is because the loading of Mg^2+^ weakens and destroys the crystal structure, enhances the crosslinking and repolymerization of the pyrolysis intermediates, and facilitates the thermal degradation process of the sample [32]. It serves as carbon fixation and encourages carbon preservation in solid form. The outcomes show that adding MgCl_2_ enhances the catalytic pyrolysis heat degradation process.

### 3.2. Effect of MgCl_2_ Loading on Product Properties

#### 3.2.1. Characterization of Carbon Materials

To investigate the effect of carbon-fixing agents on the morphology of biomass-activated carbon, electron microscopy scans were performed on CC, ZnCC, and 1% Mg-ZnCC. From Figure 2a, we can see that the surface morphology of the activated carbon produced by the hydrothermal heat of cabbage without adding an activator and carbon-fixing agent is dominated by a smooth blocky structure with few pore sizes. However, in Figure 2b,c, the significant addition of the ZnCl2 activator results in a porous and loose structure with a rough surface on the activated carbon. The electron microscopy scans on CC, ZnCC, and 1% Mg-ZnCC were carried out to examine carbon-fixing agents’ impact on biomass-activated carbon morphology. From Figure 2a, a smooth and blocky structure with small pore sizes dominates the surface morphology of the activated carbon created by the hydrothermal cabbage heat without adding an activator and carbon fixing agent. Due to the significant presence of the ZnCl_2_ activator in Figure 2b,c, the activated carbon surface appears porous, has a loose structure, and has a rough surface. The surface of the carbon material becomes loose and rough after ZnCl_2_ activation.

The surface morphology of Figure 2c was more porous and sparser, suggesting that the surface may have a more developed pore structure. Transmission electron microscopy (TEM) images of the samples were taken in order to further investigate the surface morphology of 1% Mg-ZnCC. As shown in Figure 2d, there are many black dots on the surface of the carbon material, which can be attributed to MgO particles. Additionally, MgO nanoparticles can quickly form particle agglomerates because they are a crystalline material with small particles, strong ion exchange capacity, and high surface chemical energy [33]. This phenomenon’s presence can demonstrate the effectiveness of Mg element doping. The presence of numerous white dots with an average diameter of about 11.44 nm on the surface of the thin sample material indicates the presence of numerous mesoporous structures in the carbon material. These white dots are overlapped pores or holes in the carbon material. Because the carbon material is thin in this region, the white dots were formed under transmission electron microscopy.

To investigate the effect of the activator and carbon-fixing agent on the structure of cabbage-based activated carbon, a gas adsorber was used to test the nitrogen adsorption-desorption isotherm curves of each sample. Subsequently, data processing software was used to calculate the structural parameters, including pore structure, specific surface area, pore size distribution map, and micro mesoporosity of the samples. The nitrogen adsorption-desorption curve, as shown in Figure 2e, exhibits a classic type IV isotherm with an H4 hysteresis loop, demonstrating the presence of micro- and mesoporous structures in the substance [34].

The mesoporosity ratio increased from 85.70% to 89.64%, which is much higher than previously reported cabbage-based carbon materials [35], according to the pore structure analysis (Table 1). The high mesoporosity may be caused by the stripping of hydrogen and oxygen from the feedstock by ZnCl_2_ and the generation of gas, which activates the activated carbon. MgO was embedded between carbon layers to increase the surface area and pore volume of activated carbon material, which may have contributed to the 1% Mg-ZnCC mesopore ratio. From the mesopore size distribution Figure 2f, the mesopore size is mainly distributed in the range of 2 nm–15 nm.

#### 3.2.2. Effect of MgCl_2_ Doping on Carbon Yields

The raw material is evaluated to ascertain reaction conditions, judge its value as a raw material for producing carbon, and perform other essential tasks. The C, H, O, and N content of the raw materials and carbon materials was assessed using elemental analyzers (Table 2). As can be observed from Table 2, cabbage has a high elemental content of both C and O. Following hydrothermal treatment, the O percentage significantly drops, and the C percentage increases to 66.12%. ZnCl_2_ lowers the pyrolysis temperature and contributes to the pyrolysis, which happens at 200 °C in a hydrothermal, drying environment, increasing the percentage of C, and lowering the percentage of O, most likely because it is added. As the proportion of the elements had no discernible impact on the samples, adding 1% Mg in the presence of ZnCl_2_ had minimal impact.

The solid yield can reflect the efficiency of resource utilization. More fixed elements, primarily carbon, result from high carbon yield, which is advantageous for carbon reduction. It is easy to see from Table 2 that the charcoal yield of CC is 39.08%, the charcoal yield of ZnCC is 19.26%, and the charcoal yield of 1% Mg-ZnCC is 24.81%. The reduction in ZnCC-activated carbon yield may be caused by the catalytic effect of ZnCl_2_ to break the C-O and C-C bonds in biomass and generate small molecules such as CO_2_ and methane gas or liquid substances. The addition of Mg salt leads to an increase in the charcoal yield of activated carbon.

On the one hand, under high-temperature conditions, Mg ions formed stable MgO nanoparticles that were embedded between layers of carbon. This increased the weight of activated carbon derived from cabbage and increased the carbon yield. On the other hand, Mg’s catalytic impact improved the crosslinking and repolymerization of the pyrolysis intermediates, increasing biochar yields. As can be seen in Table 2, the fixed carbon yield of 1% Mg-ZnCC is 12.14% higher than that of ZnCC. This indicates that adding Mg not only improved the charcoal yield but also significantly increased the fixed carbon yield. It is helpful to improve the fixed carbon yield and reduce carbon emissions.

#### 3.2.3. FTIR Analysis of Cabbage-Based Activated Carbon

Fourier infrared spectroscopy (FT-IR) was utilized to examine the surface functional groups of carbon materials to comprehend the evolution of functional groups in carbon materials caused by MgCl_2_. As shown in Figure 3a, the peaks at 3400 cm^−1^, 1600 cm^−1^, and 1110 cm^−1^ (Aromatic C-O-C) clearly show the surface carbon structure of biochar, which is mainly consists of chain hydrocarbons, hydroxyl groups, and aromatic rings [36]. The FT-IR spectra of ZnCC and 1%Mg-ZnCC show significant changes, indicating that the functional groups changed with adding Mg. The intensity of the O-H bond stretching vibration peak at 3400 cm^−1^ and the stretching vibration of C=C on the aromatic ring at 1600 cm^−1^ for 1% Mg-ZnCC [37] is much higher than the peaks at the corresponding positions for ZnCC, which can be attributed to the stretching vibration of the Mg-O bond [38]. For 1% Mg-ZnCC, the Mg-O-Mg (1400 cm^−1^–1500 cm^−1^) vibrations are enhanced after MgO modification, suggesting that adding MgO may provide more adsorption sites [37]. The peaks of the stretching and bending vibrations of Mg-O appear between 400 cm^−1^ and 850 cm^−1^ [37]. More interestingly, the C-H peak of 1% Mg-ZnCC at 885 cm ^−1^ is lower than that of ZnCC, possibly because the addition of Mg inhibits the deformation of C-H in the aromatic ring [39].

#### 3.2.4. XRD Analysis of Cabbage-Based Activated Carbon

In Figure 3b, the XRD spectra of biochar 1% Mg-ZnCC are displayed. It was found that there are distinct MgO diffraction peaks at 36.9°, 42.9°, 62.4°, 74.7° and 78.6° at 2θ, as well as broad spectrum diffraction peaks of amorphous carbon. This finding is in line with the findings of earlier research [27] that showed that the pyrolysis of Mg salts supported by biomass completely converted inorganic Mg salts into MgO microcrystals [40].

Additionally, the MgCl_2_ obtained from the Scherrer equation results in MgO crystals with a size of 11.5 nm [41], somewhat close to the mesopore pore size from the BET analysis in Table 1. This suggests that if too much MgCl_2_ is added, it results in the production of many MgO crystals, which may block the pore channels of the activated carbon.

#### 3.2.5. Zeta Potential

The surface charge is a crucial element influencing the effectiveness of adsorption, and it is frequently studied using the zeta potential. The Zeta potential of 1% Mg-ZnCC at pH = 2 ~ 11 is tested and plotted. According to Figure 3c, this material’s zero-point charge (pH zpc) value is 6.6, near 7, showing that 1% Mg-ZnCC can adjust to sizable pH variations in solution. When pH < pH zpc, the 1%Mg-ZnCC surface is positively charged, favoring the adsorption of anionic contaminants; when pH > pH zpc, the 1%Mg-ZnCC surface is negatively charged, favoring the adsorption of cationic contaminants [42,43]. The effect of pH on the adsorption capacity of 1%Mg-ZnCC is shown in Figure 3f As the pH value increases, the adsorption capacity initially increases and then decreases, reaching the maximum value at pH 7. Previous studies have shown that TC is an amphoteric organic substance, which is positively charged when the pH of the solution is less than 3.3 and negatively charged when the pH is above 7.7 [44,45]. Therefore, too low (pH < 3.3) or too high pH value (pH > 7.7) causes electrostatic repulsion between 1%Mg-ZnCC and TC, which is not conducive to TC adsorption. In addition, there may be chemisorption in the adsorption of TC [46], resulting in adsorption results that cannot fully conform to the law of electrostatic adsorption.

#### 3.2.6. XPS

XPS is commonly used to determine the valence and surface elemental composition of 1% Mg-ZnCC. XPS was used to examine the chemical valence and functional group binding of Mg 1 s (Figure 3d). A single peak at the binding energy of 1304.4 eV was observed, which was attributed to the Mg-O bond, indicating the successful loading of MgO [47]. The XPS peak shift analysis demonstrated that Mg was bound to oxygen.

### 3.3. Contaminant Adsorption

The best adsorbent for removing contaminants from water is activated carbon (AC) [48,49,50]. For instance, low-cost carbon-based adsorbents have been created using rice straw [51], coconut shells [52], coffee [53], fruit kernels [54], nut shells [55], grape seed [56], corn cobs [57], and corn stalks [58]. Extracting low-cost activated carbon from waste is more economically possible than using commercial coal-based activated carbon for wastewater treatment [59,60]. The adsorption of methylene blue solution with an initial concentration of 100 mg·L^−1^, acidic magenta solution, potassium dichromate solution, and tetracycline solution with an initial concentration of 50 mg·L^−1^ was examined while comparing 1% Mg-ZnCC and ZnCC (Figure 3e). Overall, the adsorption capacity of 1% Mg-ZnCC was more substantial than that of ZnCC for all four pollutants, with increases of 8.71 mg, 37.15 mg, 117.68 mg and 3.53 mg, respectively. However, there was little change in the 1% Mg addition’s impact on the tetracycline adsorption. This may be because MgCl_2_ enhances the crosslinking and repolymerization of pyrolysis intermediates during the pyrolysis process. The MgO particles generated by pyrolysis are embedded between carbon layers, increasing the surface area and void structure of activated carbon, resulting in increased adsorption properties after adding MgCl_2_.

## 4. Conclusions

The sample’s crystal structure was degraded and fragmented by adding Mg, which encouraged its thermal degradation process and decreased its maximum pyrolysis rate [32]. By lowering the material’s pyrolysis temperature and reducing the release of carbonaceous gaseous material, the catalytic impact of Mg improved the crosslinking and polymerization reactions of the pyrolysis intermediates, increasing biochar yield [61]. Due to the presence of the element Mg, which prevented the weaker bonded substituent from breaking, the addition of Mg weakened the carbonyl C=O. Additionally, the magnesium element reduced the dehydroxylation reaction and increased the strength of the C-H signal. Mesoporosity and specific surface area rose due to the Mg-induced increase in active sites, and the mesoporosity ratio increased from 85.70% to 89.64%. Compared to ZnCC, the fixed carbon output of 1% Mg-ZnCC increased by 12.14 per cent. Characterization showed that Mg was successfully incorporated as MgO into the biomass-activated carbon, promoting carbon preservation in solid form and serving as a carbon fixation agent. The growth of its pore structure was aided by the addition of Mg salts, which also improved the activated carbon’s capacity for adsorption and altered the morphological traits and chemical composition of the biochar material. This result increases the adsorption capacity by 8.71 mg, 37.15 mg, 117.68 mg and 3.53 mg for methylene blue solution with the initial concentration of 100 mg·L^−1^, acidic magenta solution, potassium dichromate solution and tetracycline solution with the initial concentration of 50 mg·L^−1^, respectively.

## Figures and Tables

**Figure 1 bioengineering-10-00836-f001:**
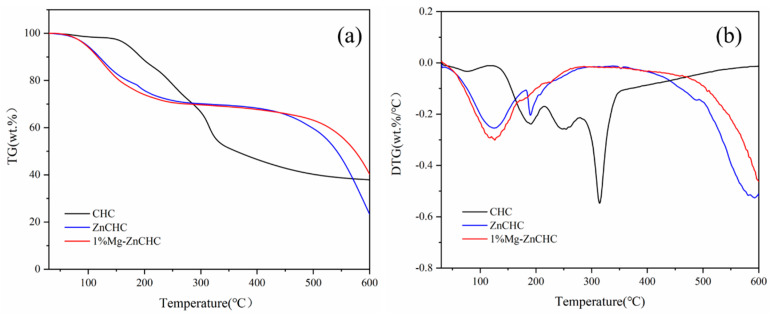
(**a**) TG curves of CHC, ZnCHC and 1% Mg-ZnCHC, (**b**) DTG curves of CHC, ZnCHC and 1% Mg-ZnCHC.

**Figure 2 bioengineering-10-00836-f002:**
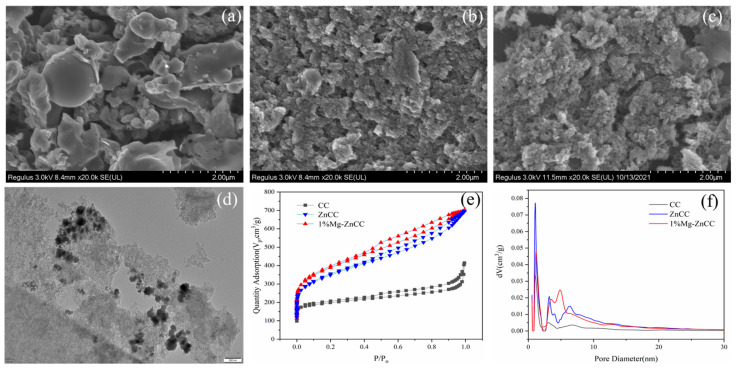
SEM images of (**a**) CC, (**b**) ZnCC, (**c**) 1%Mg-ZnCC. (**d**) TEM of 1%Mg-ZnCC. (**e**) N_2_ adsorption-desorption curve. (**f**) Pore size distribution of carbon material.

**Figure 3 bioengineering-10-00836-f003:**
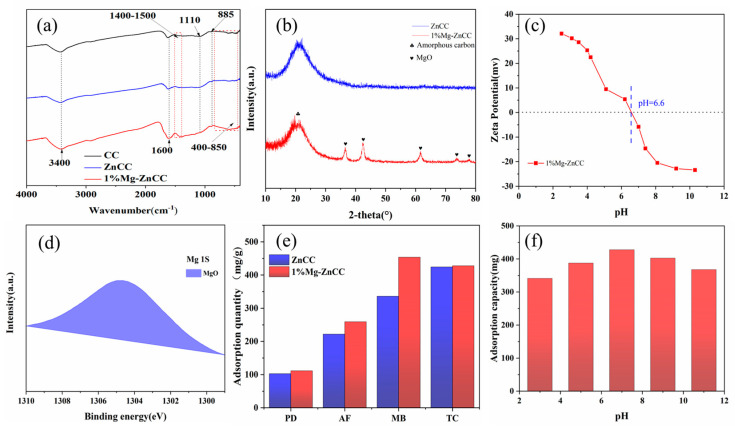
(**a**) FT-IR analysis spectrum of activated carbon. (**b**) XRD. (**c**) Zeta potential. (**d**) XPS analysis of activated carbon. (**e**) Pollutant adsorption comparison (C_0_ = 100 mg/L; adsorbent dose = 0.01 g; V_0_ = 50 mL). (**f**) effect of pH on TC removal capacity.

**Table 1 bioengineering-10-00836-t001:** Specific surface and pore size distribution of cabbage-based activated carbon.

Name	Specific Surfaceeik (m^2^/g)			Aperture (nm)	Hole Capacity (cc/g)		
	Microporous	External Surface	Total		Microporous	Mesopore	Total
CC	262.165	200.08	462.245	5.67	0.114	0.418	0.5326
ZnCC	289.967	738.149	1028.12	10.61	0.133	0.7972	0.9302
1%Mg-ZnCC	229.193	855.444	1084.64	11.44	0.106	0.9174	1.0234

**Table 2 bioengineering-10-00836-t002:** Elemental analysis.

Name	C [%]	H [%]	O [%]	N [%]	Charcoal Yield	Carbon Yield
Cabbage	31.58	5.203	38.274	4.54		
CC	66.12	1.211	14.7033	5.16	39.08	81.81
ZnCC	76.14	2.037	7.9948	3.54	19.26	46.44
1%Mg-ZnCC	74.58	3.247	9.0961	2.77	24.81	58.58

Charcoal Yield: Ratio of char material to biomass weight after biomass pyrolysis. Carbon Yield: Ratio of carbon in biomass char material to carbon in biomass.

## Data Availability

Not applicable.

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
