# Peer review of "Effect of MgCl2 Loading on the Yield and Performance of Cabbage-Based Biochar"

_bioengineering, 2023, doi:10.3390/bioengineering10070836_

Round 1

Reviewer 1 Report

The current study entitled “Effect of MgCl2 loading on the yield and performance of cabbage-based biochar”. The present manuscript is written well and it has the potential for publication in Bio-engineering journal.

The abstract section is written well, however, authors I suggest the authors to add a rationale (1-3 lines) at the start of the abstract section.

At the end of the abstract section, the authors must also add a strong conclusion.

Though introduction section is written well, however, authors can add a hypothesis before the objectives in revised introduction section.

The discussion section must be enriched with more logical reasoning.

There are some grammatical mistakes in the manuscript, please revisit the manuscript and remove all these grammatical mistakes.

The quality of the English language needs moderate changes. 

Author Response

The current study entitled “Effect of MgCl2 loading on the yield and performance of cabbage-based biochar”. The present manuscript is written well and it has the potential for publication in Bio-engineering journal.

Comment 1.

The abstract section is written well, however, authors I suggest the authors to add a rationale (1-3 lines) at the start of the abstract section.

Response:

Thank you for your suggestion. We have added a rationale at the start of the abstract section.

Comment 2.

At the end of the abstract section, the authors must also add a strong conclusion.

Response:

Thank you for your suggestion, we have improved the Abstract part.

Converting more CO2 absorbed by plant photosynthesis into biomass activated carbon is an effective way to reduce carbon emissions. In this study, we used a one-step preparation of biomass-activated carbon loaded with MgO nanoparticles to investigate the effect of Mg loading on the catalytic pyrolysis process. The influences of magnesium loading on biochar yield and fixed carbon production were assessed.1% Mg addition weakened the carbonyl C=O, inhibited the dehydroxylation reaction, enhanced the C-H signal strength, and the formation of MgO inhibited the weaker bound substituent breakage. Additionally, the addition of magnesium altered the morphological features and chemical composition of the biochar material, and also increased the activated carbon mesoporosity by 3.94%, biochar yield by 5.55% and fixed carbon yield by 12.14%. The addition of 1% Mg increased the adsorption capacity of the activated carbon to potassium dichromate, acid magenta, methylene blue, and tetracycline effluents by 8.71 mg, 37.15 mg, 117.68 mg and 3.53 mg, respectively. The results showed that MgCl2 played a significant role in promoting the thermal degradation of biomass and improving the solid yield and adsorption performance of activated carbon.

Comment 3.

Though introduction section is written well, however, authors can add a hypothesis before the objectives in revised introduction section. thank you.

Response:

According to your suggestion, we have added a hypothesis before the objectives in introduction section. thank you.

Comment 4.

The discussion section must be enriched with more logical reasoning.

Response:

Thank you for your suggestion, we have enriched logical reasoning, such as 3.1, 3.2.1, 3.2.3, etc.

Comment 5.

There are some grammatical mistakes in the manuscript, please revisit the manuscript and remove all these grammatical mistakes.

Response:

Thank you for your advice. We have carefully checked and corrected the grammar and spelling errors in the manuscript.

Reviewer 2 Report

The authors have presented a detailed study of the effect of MgCl2 on the yield and performance of biochar. However, the manuscript requires following major revisions prior to acceptance.

1.      Please add a brief discussion on how MgCl2 improves the performance of biochar in the introduction section and why it is chosen for modification of biochar. Also, cite some relevant studies.

2.      Please enhance the literature review by citing : https://doi.org/10.3390/su142114571, https://doi.org/10.3390/w15030394 and more relevant articles.

3.      In section 3.2.2, please explain the reasons for the reduction of charcoal and carbon yield for ZnCC compared to CC.

4.      Section 3.3 should be elaborated. The results obtained from the characterization of biochar should be correlated with the adsorption performance.

5.      In section 2.4.5, there is no mention of X-ray Photoelectron Spectroscopy (XPS). Please include that. Furthermore, there are some mistakes in sentence formation in that section. Please improve.

6.      In section 3.2.5, pH PZC is mentioned. The author must include a brief discussion about PZC and abbreviation before abruptly using it in the manuscript.

7.      The effect of pH on the adsorption of pollutants could be studied to validate the zeta potential obtained in section 3.2.5.

8.      Please adjust the subscripts and superscripts where it is necessary throughout the manuscript.

9.      There are several spelling mistakes and grammatical errors throughout the manuscript. Please thoroughly check and correct those mistakes.

Minor English edit including syntax error, spelling mistakes, sentence structure.

Author Response

Dear Editor:

Enclosed is our revised manuscript entitled “Effect of MgCl2 loading on the yield and performance of cabbage-based biochar” which is being submitted for publication in Bioengineering.

We truly appreciate the reviewers for their very professional and detailed suggestions, which help us to improve this work. Revisions have been made throughout the manuscript following the suggestions from the reviewers. The mention of X-ray Photoelectron Spectroscopy (XPS) was added. The effects of PH on the adsorption of TC by 1% Mg-ZnCC was explored. Some explanations have been added to the manuscript.

The clarity of the paper and the scientific study of MgCl2 loading on improving the yield and adsorption properties of biomass activated carbon was improved after these revisions and we believe that the new version of the manuscript is acceptable for publication in Bioengineering.

Thanks a lot!

Sincerely yours,

Lu Liu

School of Engineering

 Anhui Agricultural University

Hefei, Anhui, 230036, PR, China

Round 2

Reviewer 2 Report

Manuscript ID: bioengineering-2448354

Comments

1.      “For batch adsorption tests, ten mg of 1% Mg-ZnCC and ZnCC were accurately weighed. “

Please replace ten to 10.

2.      “And adding them to 50 mL of methylene blue solution, acidic magenta solution, potassium dichromate solution, and tetracycline solution with an initial concentration of 100 mg·L-1, respectively.” The sentence is not complete. 

3.      Provide a bit detail on adsorption experiment like pH, temperature, time for equilibrium attaining, etc.

4.      How about adsorption isotherm? It is better if the authors can provide adsorption capacity instead of showing % removal.

5.      In 3.2.6, provide a bit details on XPS results.  How does it relate with the adsorption experimental results?

6.      English correction is required.

Minor English editing is required.

Author Response

Dear Editor:

Enclosed is our revised manuscript entitled “Effect of MgCl2 loading on the yield and performance of cabbage-based biochar” which is being submitted for publication in Bioengineering.

We truly appreciate the reviewers for their very professional and detailed suggestions, which help us to improve this work. Revisions have been made throughout the manuscript following the suggestions from the reviewers. A bit detail on adsorption experiment was added. Some explanations have been added to the manuscript.

The clarity of the paper and the scientific study of MgCl2 loading on improving the yield and adsorption properties of biomass activated carbon was improved after these revisions and we believe that the new version of the manuscript is acceptable for publication in Bioengineering.

Thanks a lot!

Sincerely yours,

Lu Liu

School of Engineering

Anhui Agricultural University

Hefei, Anhui, 230036, PR, China

Referee: 1

Comment 1.

 “For batch adsorption tests, ten mg of 1% Mg-ZnCC and ZnCC were accurately weighed. “

Please replace ten to 10.

Response:

Thank you for your suggestion. We have replaced ten to 10.

Comment 2.

 “And adding them to 50 mL of methylene blue solution, acidic magenta solution, potassium dichromate solution, and tetracycline solution with an initial concentration of 100 mg·L-1, respectively.” The sentence is not complete.

Response:

Thank you for your suggestion, we have perfected the sentence, The relevant information has been added to the 2.5.

Comment 3.

Provide a bit detail on adsorption experiment like pH, temperature, time for equilibrium attaining, etc.

Response:

Thank you for your suggestion, we have perfected the sentence, such as 2.5.

Comment 4.

How about adsorption isotherm? It is better if the authors can provide adsorption capacity instead of showing % removal.

Response:

Thank you for your very professional suggestions. This work mainly studied the carbon yield effect of MgCl2 and the changes in the adsorption capacity of cabbage-based activated carbon to several common pollutants before and after the addition of MgCl2, as shown in 3.3 and Figure 3, which reflects that the addition of MgCl2 can promote the adsorption of pollutants by activated carbon. As for the maximum adsorption amount of each pollutant of biochar, we will determine it through adsorption isotherm experiment and fitting of Langmuir, Freundlich, Dubinin-Radushkevich and Temkin and other methods in the later stage.

Comment 5.

In 3.2.6, provide a bit details on XPS results.  How does it relate with the adsorption experimental results?

Response:

The XPS analysis in 3.2.6 is mainly to prove that MgCl2 participates in the pyrolysis reaction and generates new MgO particles in the carbon material. After MgCl2 was added, the adsorption capacity of pollutants on cabbage-based activated carbon increased. We speculated that MgO nanoparticles doped between carbon layers increased the pore structure and specific surface of activated carbon, thus increasing the adsorption performance. Due to the small amount of MgCl2 added, the cabbage-based activated carbon may have complex chemical or physical adsorption effects on different pollutants. If possible, we will conduct specific adsorption mechanism research in the later stage. Thank you.

Comment 6.

English correction is required.

Response:

Thank you for your advice. We have corrected the English expression of the work.